behaviour

cooperation, coordination, cognition, communication, bottlenose dolphin

**Authors for correspondence:**
Stephanie L. King
e-mail: stephanie.king@bristol.ac.uk
Kelly Jaakkola
e-mail: kelly@dolphins.org

# Evidence that bottlenose dolphins can communicate with vocal signals to solve a cooperative task

Stephanie L. King[1], Emily Guarino[2], Katy Donegan[2], Christina McMullen[2] and Kelly Jaakkola[2]

[1]School of Biological Sciences, University of Bristol, Bristol BS8 1TQ, UK
[2]Dolphin Research Center, 58901 Overseas Highway, Grassy Key, FL 33050, USA

 SLK, 0000-0003-2293-9185; KJ, 0000-0002-4113-748X

Cooperation experiments have long been used to explore the cognition underlying animals' coordination towards a shared goal. While the ability to understand the need for a partner in a cooperative task has been demonstrated in a number of species, there has been far less focus on cooperation experiments that address the role of communication. In humans, cooperative efforts can be enhanced by physical synchrony, and coordination problems can be solved using spoken language. Indeed, human children adapt to complex coordination problems by communicating with vocal signals. Here, we investigate whether bottlenose dolphins can use vocal signals to coordinate their behaviour in a cooperative button-pressing task. The two dolphin dyads used in this study were significantly more likely to cooperate successfully when they used whistles prior to pressing their buttons, with whistling leading to shorter button press intervals and more successful trials. Whistle timing was important as the dolphins were significantly more likely to succeed if they pushed their buttons together after the last whistle, rather than pushing independently of whistle production. Bottlenose dolphins are well known for cooperating extensively in the wild, and while it remains to be seen how wild dolphins use communication to coordinate cooperation, our results reveal that at least some dolphins are capable of using vocal signals to facilitate the successful execution of coordinated, cooperative actions.

## 1. Introduction

Many animals are known to work together to perform cooperative tasks that require individuals to coordinate their behaviour in

order to succeed. Examples can be found across the animal kingdom, where individuals have been shown to work together in order to improve foraging efficiency (e.g. ants [1], wild-dogs [2], sailfish [3] and humpback whales [4]), dominate in inter-group contests (e.g. dolphins [5], lions [6], chimpanzees [7] and mongooses [8]) and enhance reproductive success (e.g. baboons [9] and dolphins [5]). Such tasks require the coordinated performance of two or more individuals and, in many animals, communication appears to facilitate this coordination. However, our understanding of how individuals use and share information during such coordinated group tasks remains limited [10].

Cooperation experiments have traditionally been used to explore the cognition underlying coordination towards a shared goal [10–12]. The cooperative rope pulling paradigm widely used in these studies requires two animals to coordinate their efforts by each pulling a rope in the same direction in order to gain a food reward [13]. This experimental design has also been adapted such that individuals coordinate the pulling of handles or the pushing of buttons rather than the pulling of ropes [14,15]. An important component of all these experiments is testing whether individuals will wait for their partner before manipulating the device, demonstrating that they understand the cooperative nature of the task. Experiments such as these have been used to show that some species, including chimpanzees [16–20], orangutans [21], capuchins [15,22], elephants [23], wolves [24], hyaenas [25], keas [26], bottlenose dolphins [14] and peach-front conures [27] understand the need for a partner, whereas other species, including otters [28], rooks [29], ravens [30] and African grey parrots [31] do not. Yet there has been less focus on cooperation experiments that address the role of communication [10]. While language facilitates collective action in humans [32], our knowledge of the role that communication plays in facilitating cooperative endeavours in non-human animals remains limited.

Recent work has shown that great apes can use communication, specifically visual signals, to facilitate task success [33,34]. This is in contrast to humans, however, who can use vocal communication. For example, a study comparing coordination strategies in chimpanzees and human children revealed greater success in humans because they switched to vocal communication when vision was restricted, which facilitated their continued success [35].

Bottlenose dolphins are well known for cooperating extensively in the wild, including feeding strategies that require coordinated action [36,37] and the formation of long-term cooperative alliances, where males work together to sequester individual females [5,38]. Here, we investigated whether bottlenose dolphins use vocal signals to coordinate their behaviour in a cooperative task. During this task, members of a dyad are required to swim across a lagoon and each press their own underwater button within 1 s of each other. If there was more than a 1 s delay between the two button presses, the dyad was unsuccessful (see methods for full description). We have previously shown that dolphin dyads are able to work together with extreme precision during this task, even when they have to wait for their partner [14]. The fact that this experimental design requires tight behavioural synchronization eliminates the possibility that success could be achieved from mechanisms such as response facilitation or responding to some environmental cue [14]. Thus, consistent success during this task is suggestive of active coordination [12]. Using this experimental design, the current study explored how two dolphin dyads (a male dyad and a female dyad) actively used vocal signals during trials that increased in task difficulty, ranging from partners being sent together, to progressive delays between partners, to partners being unable to see one another when they pressed their buttons. We recorded the physical and vocal behaviour of the dyads during these tasks, using a four-element hydrophone array to localize vocalizations to determine how individuals used vocal signals when working together.

# 2. Methods

## 2.1. Subjects

Experiments were conducted at Dolphin Research Center (DRC) in Grassy Key, Florida between January 2018 and August 2019. The subjects were four common bottlenose dolphins: Aleta (female, 33 years old) and Calusa (female, 17 years old) who formed dyad 1; and Delta (male, 9 years old) and Reese (male, 7 years old) who formed dyad 2. Dyad 1 participated in the original study [14] and dyad 2 were new to this study. All four animals were born at DRC and lived in natural seawater lagoons (ranging from 344 to 537 m$^2$) with depth dependent on tide (4.5–5.5 m). The members of each dyad had lived together at various points throughout their lives and lived together during the study. All dolphins at DRC voluntarily participate in three to five positive reinforcement training sessions daily that may include husbandry, behavioural training, play sessions, public interactions with trainers and guests,

and research. Behavioural training includes solo and tandem physical behaviours (e.g. asking two dolphins to dive together), as well as conceptual behaviours (e.g. repeat, imitate, do something new). Throughout the study, the dolphins were fed according to their normal daily routine, which typically included capelin, herring, smelt and squid three to five times per day, approximately 20–33% of which they received during each experimental session (up to two sessions per day).

## 2.2. Cooperative task apparatus and procedure

A full description of the cooperative task is provided in Jaakkola *et al*. [14]. During this task, members of a dyad were required to swim across a lagoon (approx. 11 m) and each press their own underwater button simultaneously (within a 1 s time window), whether sent together or with a delay between partners of 1–20 s. The buttons were connected via a computer (Raspberry Pi Model 3 B+), which in turn was attached to an underwater speaker (University Sound UW-30). If the buttons were pressed within this 1 s time interval, the computer automatically played a 'success' sound (i.e. a trainer's whistle) and the dolphins returned to the trainers across the lagoon for positive reinforcement of fish and social interaction. If there was more than a 1 s delay between the dolphins' button presses, the computer played a 'failure' sound, and no reinforcement was given. For each trial, only the first press of each button was relevant, and it was impossible for dolphins to succeed by repeatedly pushing their buttons. The computer automatically recorded the following parameters for each trial: time between button presses (accurate to 0.01 s), which button was pressed first, and whether the trial outcome was a success or failure. A Canon Vixia HF R50 video camera positioned across the lagoon from the apparatus and a GoPro Hero5 positioned above the apparatus were used to record the trials.

Because this task included no perceptible causality by which the dolphins could deduce the need to work together, each dolphin dyad had to pass a series of trial phases in order to learn the task and then demonstrate they understood the cooperative nature of the task (phases 0–4 in our original study [14]). For this study, we ran additional phases to determine how restricted vision might influence the dyad's vocal behaviour (phases 5–8). A full summary of all the trial phases, and the number of trials conducted for each phase, are provided in table 1.

### 2.2.1. Simultaneous release and delayed release conditions

Initially, both individuals in each dyad were sent together (phase 0: simultaneous release) until they had passed a criterion of two sessions of 20 trials each with an 80% success rate. Incremental delays were then introduced where one individual was released first, followed by a short delay until the second member of the dyad was released (phase 1: incremental delays of 1–5 s). Each member of the dyad needed to pass three trials in a row (by waiting for their partner) before moving on to the next stage (123 trials for the female dyad; 115 trials for the male dyad). This was followed by a session of approximately 40 trials per dyad where we tested them with randomized delays (phase 2: randomized delays of 0–5 s). We then increased the delay times (phase 3: incremental delays of 8–20 s) where each member of the dyad needed to pass three trials in a row (199 trials for the female dyad; 58 trials for the male dyad). This was followed by another session of approximately 40 trials per dyad where we tested them with the longer randomized delays (phase 4: randomized delays of 1–20 s). During all these trials, the buttons were located next to each other on the front of the dock (position A, figure 1).

### 2.2.2. Restricted vision trials

We then moved the buttons to opposite sides of the dock to ensure the individuals could not rely on vision to coordinate their button press (position B, figure 1). We tested both dyads in these *restricted vision* trials, where members of the dyad were released from three different locations in the lagoon, sometimes starting at the same location and sometimes starting from different locations (phase 5: restricted vision 1—simultaneous release). Each dyad needed to pass three trials in a row at each configuration of locations (55 trials for the female dyad; 48 trials for the male dyad). Because the male dyad excelled at this task, we conducted additional restricted vision trials to explore their abilities. First, we introduced randomized delays where members of the dyad were released from one of the three previous locations (either both from the same location or from different locations). Members of the dyad were either released together or with a 5 or 10 s delay (phase 6: restricted vision 1—randomized delays of 0, 5 or 10 s) for a total of 135 trials. Finally, to make the task even more difficult for the male dyad, we moved the buttons to different parts of the lagoon (position C and D, figure 1). Both individuals were sent from the original location (as per phases

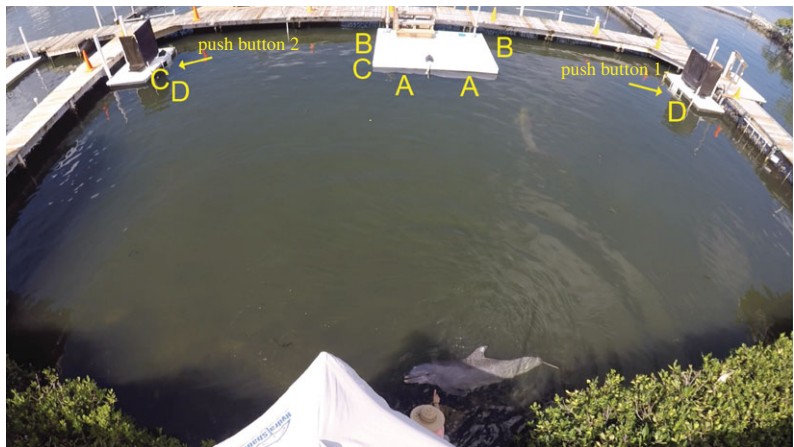

**Figure 1.** View of the experimental set-up for the different trial phases. (A) Buttons positioned next to each other on the dock for phases 0–4. (B) Buttons positioned facing away from each other on the dock for phases 5 and 6. (C) Buttons positioned half-way across the lagoon for phase 7. (D) buttons positioned on either side of the lagoon for phase 8.

**Table 1.** Summary of all trial phases.

|  | trial type | button location | distance between buttons (m) | criterion to pass |
|---|---|---|---|---|
| Phase 0 | simultaneous release | A | 2.6 | 8 out of 10 over two sessions (80%) |
| Phase 1 | incremental delays (1–5 s) | A | 2.6 | 3 in a row |
| Phase 2 | randomized delays (simultaneous − 5 s) | A | 2.6 | 16 out of 20 in a single session (80%) |
| Phase 3 | incremental delays (8–20 s) | A | 2.6 | 3 in a row |
| Phase 4 | randomized delays (1–20 s) | A | 2.6 | test (20 trials per dolphin) |
| Phase 5 | restricted vision 1: simultaneous release | B | 3.3 | 3 in a row |
| Phase 6[a] | restricted vision 1: randomized delays (simultaneous, 5 s and 10 s) | B | 3.3 | test (~40 trials per location) |
| Phase 7[a] | restricted vision 2: randomized delays (simultaneous, 5 s and 10 s) | C | 8 | test (~40 trials) |
| Phase 8[a] | restricted vision 3: randomized delays (simultaneous, 5 s and 10 s) | D | 17.65 | test (~40 trials) |

[a]Only the male dyad took part in these phases as they excelled in the initial trial phases.

0–4) and we included randomized delays for approximately 40 trials for the mid-lagoon set-up (phase 7: restricted vision 2—randomized delays of 0, 5 or 10 s; position C, figure 1) and randomized delays for approximately 40 trials for the full lagoon set-up (phase 8: restricted vision 3—randomized delays of 0, 5 or 10 s; position D, figure 1). A full summary of all the trial phases are provided in table 1, and the number of trials and success rate are provided in electronic supplementary material, table S1. Videos of some delayed release trials are provided at the Dryad Digital Repository [39].

## 2.3. Acoustic data collection

During all trials, vocalizations were recorded using a hydrophone array consisting of four HTI-96 MIN series (flat frequency response: 0.002–30 kHz ± 1 dB) onto a TASCAM DR-680 MKII multi-track recorder at a sampling rate of 96 kHz. For the female dyad, we placed one hydrophone in each corner of the lagoon in an approximate rectangular formation, and distances between hydrophones ranged from approximately 7.8 to 20.8 m. Acoustic localization was used to confirm that the whistles came from within the lagoon

and from the focal dyad. Localization error was calculated by asking a dolphin to whistle on signal at a known location at the dock, and then using custom-written MATLAB routines to calculate two-dimensionally averaged MINNA (minimum number of receiver array) localizations using the methods described in Wahlberg *et al.* [40] and Schulz *et al.* [41], and was found to be 1.3 m from the true location of the calling animal [42]. For the male dyad, we modified the array design and moved the array to the dock, where each hydrophone was placed at the corner of the dock in an approximate rectangular formation (3.3 by 3.1 m) during phases 0–6. This did not change our ability to record and localize whistles, but this configuration was logistically easier to deploy. We used the localization bearing (corroborated with amplitude differences between hydrophones) to confirm caller identification. We used the video footage above the apparatus to confirm which dolphin was pushing each button. During phases 7 and 8, the array was 8 by 1.9 m and 17.65 by 1.9 m, respectively. Images of all experimental set-ups are provided in figure 1. Spectrograms were inspected in Adobe Audition CC v. 2017.0.2 (Adobe Systems) for instances of whistle production between the start of each trial (when the dolphins are given their hand signal to start the trial) and the end of each trial (when the success or failure sound is played). Whistles were identified either as signature whistles (individual identity signals [43,44]) or non-signature whistle types. Prior to the start of this study, signature whistles were identified for three out of four dolphins by recording the most common whistle type produced by each individual when isolated [42,43]. For the fourth dolphin (Reese), the SIGID (signature identification) method was used to identify his signature whistle [45]. The number of whistles produced during each trial was recorded, and only whistles localized to the focal dyad, or confirmed as their signature whistle, were used in further analysis.

## 2.4. Video coding

We used the event logging software BORIS [46] to code the behaviour of individual dolphins during the trials in phases 1–5 for both dyads. We coded from the start of each trial (when the dolphins are given their hand signal to start the trial) to the end of each trial (when the success or failure sound is played). During the trial, we coded for the duration of the following behaviours (i) *waiting*—this is recorded when a dolphin's tail stops actively propelling them forward for 2 s and a waiting period begins (regardless of whether that waiting occurred at the button, near the delayed dolphin or somewhere in between); (ii) *swimming together*—when both dolphins have the same body orientation and are swimming within one body length of each other. One observer coded all the videos for both dyads. A second observer independently coded 20% of these same trials, which were chosen at random. We then conducted an inter-reliability analysis using the intraclass correlation coefficient (ICC) for two-way models, using the *irr* package in R 3.4.4 (R project for statistical computing; GNU project). We found excellent agreement between our two observers (ICC = 0.995, $p < 0.0001$, CI = 0.994–0.996).

## 2.5. Statistical analysis

All statistical procedures were conducted in R 3.4.4 (R project for statistical computing; GNU project). Given that sample sizes represent repeated trials with two dyads, it is not possible to generalize our findings beyond the dyads. However, to explore how the vocal behaviour of each dyad influenced their success we performed the following statistical comparisons. First, to explore if the presence or the absence of whistles predicted trial outcome, we built a generalized linear model with a binomial family for each dyad (whistle occurrence model). Our response variable was the trial outcome (failure or success) and whistle occurrence (binary) was modelled as an explanatory variable. We then tested whether whistle occurrence influenced the precision of the button press by building a linear model. Our response variable was the time between both individuals pressing their button (log transformed to achieve normality), and our explanatory variable was whistle occurrence (binary).

To explore if the number of whistles produced during a trial predicted trial outcome, we built a generalized linear model with a binomial family for each dyad (whistle frequency model). Our response variable was the trial outcome (failure or success) and number of whistles (i.e. trials with no whistles were not included) was modelled as an explanatory factor variable. To explore if the relationship between the number of whistles produced during a trial and trial outcome varied with phase, we built a generalized linear model with a binomial family, where our response variable was the trial outcome (failure or success) and trial phase was modelled as an explanatory factor variable with the number of whistles nested within trial phase. For all models, we used ANOVA (car package in R) to test whether the model explained significantly more variance than the null model (electronic supplementary material, table S2).

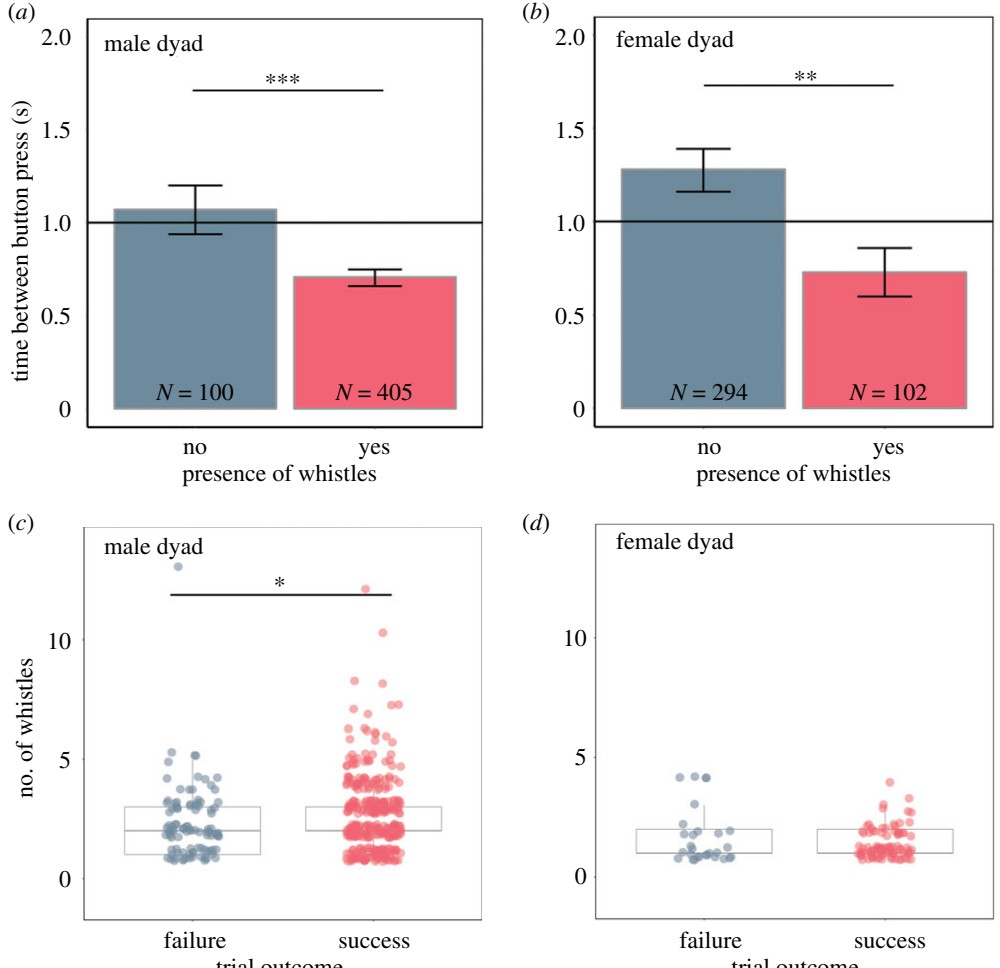

**Figure 2.** Vocal behaviour by each dyad during the cooperative task trials. The relationship between whistle absence (blue bars) and presence (red bars) and the mean time between button presses with standard errors for the male dyad (*a*) and female dyad (*b*) in trials where both animals pressed their button. Sample size (*N*) represents the number of trials per dyad with or without whistles. The horizontal line at 1 s represents the cut-off for a successful trial. The number of whistles (in trials with at least one whistle) for unsuccessful (blue dots) and successful trials (red dots) for the male dyad (*c*) and female dyad (*d*). Asterisks denote statistical significance (*** $p < 0.001$, ** $0.001 < p < 0.01$, * $0.01 < p < 0.05$).

## 3. Results

We examined the effect of whistles in two ways: we first assessed whether the presence of whistles influenced trial success; and then, for those trials with whistles, whether the number of whistles affected trial success. For both dolphin dyads, the time between their button presses was significantly shorter in trials where they whistled (male dyad estimate: −0.44, $t = -3.6$, $p = 0.0002$, figure 2*a*; female dyad estimate: −0.36, $t = -2.3$, $p = 0.01$, figure 2*b*), with whistle presence leading to significantly more successes (male dyad: $p < 0.0001$; female dyad: $p = 0.0008$), i.e. the dolphins were more likely to press their buttons within 1 s of each other if they whistled than if they did not. In general, the males were more vocal than the females (Welch's *t*-test: $t = 20.58$, d.f. = 750, $p < 0.0001$), with the male dyad producing, on average, two whistles per trial and the female dyad producing, on average, 0.37 whistles per trial. The proportion of trials that contained whistles, as well as whistle rates, remained relatively constant for the male dyad but increased across trial phases for the female dyad (electronic supplementary material, table S1 and figure S1). When we considered just those trials that included at least one whistle, more whistles also led to significantly more successful trials for the male dyad but not for the females (male dyad estimate: 0.2, $z = 2.3$, $p = 0.02$, figure 2*c*; female dyad estimate: −0.42, $z = -1.67$, $p = 0.09$, figure 2*d*). Furthermore, for both dyads, significantly more whistles were produced in successful trials compared with failed trials for those trial phases that the dolphins found particularly challenging (as measured by success rates below the mean; see electronic supplementary material, tables S1–S3).

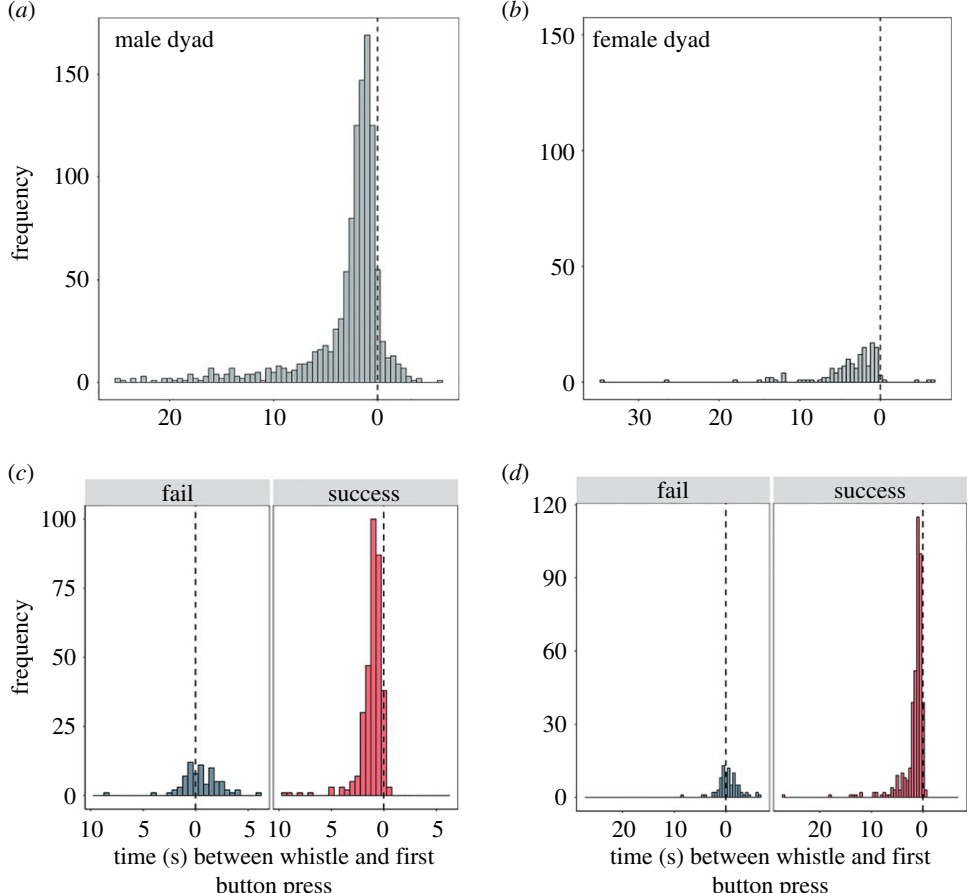

**Figure 3.** Whistle timing across trials in relation to the first button press. For all trials, the time between all whistles produced and the first button press (represented by the vertical dashed line at 0 s) for the male dyad (trial phases 0–8) (a) and female dyad (trial phases 0–5) (b). For both failed and successful trials, the time between the last whistle produced in the trial and the first button press (represented by the vertical dashed line at 0 s) for the male dyad (c) and female dyad (d). For (c,d), we excluded failures in which only one animal pressed their button (female dyad = 82 trials; male dyad = 38 trials) because we wanted to assess how whistle timing influenced coordination between the dyad, not relevant in cases in which the dolphins did not attempt to coordinate (i.e. when only one individual pushed their button).

Next, we examined the relationship between the timing of whistles and button presses. Whistles tended to occur towards the end of the trial just before the button press (figure 3a,b). For both dyads, the timing between the last whistle produced and the first button press showed a similar relation to success (figure 3c,d). Specifically, if the first button push was after the last whistle, then the dyad was significantly more likely to be successful than if they pushed before the last whistle (male dyad $X^2$ (1, $N = 405$) = 149.17, $p =$ < 0.0001, figure 3c). For the female dyad, we used a Fisher exact test because some of the expected cell frequencies were too small for a $X^2$ test. This was also significant ($p < 0.0001$, figure 3d). This further supports the idea that vocalizations played a role in coordinating the timing of button presses for these two dyads.

In addition to differences in vocal rates, the dyads also showed differences in their physical behaviour, suggesting different behavioural strategies for coordinating with their partner. Specifically, the female dyad spent a significantly greater proportion of their time waiting in place for their partner than did the male dyad (Welch's $t$-test: $t = 6.3$, d.f. = 766, $p < 0.0001$) and a significantly greater proportion of their time swimming together when approaching the buttons (Welch's $t$-test: $t = -3.2$, d.f. = 766, $p = 0.001$). The male dyad, by contrast, tended to explore other areas of the lagoon until their partner was released, meeting up at the buttons prior to pressing.

In order to investigate whistle coordination in more detail, we identified which members of the dyads were vocalizing to determine whether they were exchanging information prior to a button press. We defined a vocal exchange as whistles produced by each member of the dyad that occurred within 1 s of each other, in line with other studies on the time intervals between counter-calling pairs of

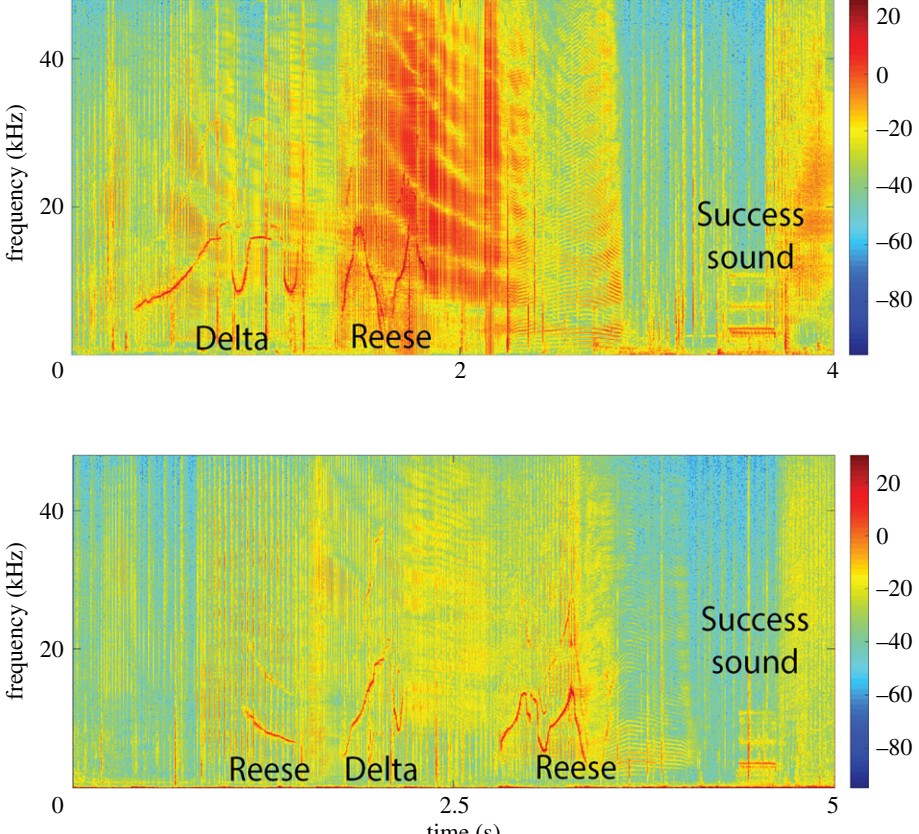

**Figure 4.** Vocal exchanges during cooperative tasks. Spectrograms showing examples of vocal exchanges for the male dyad (sampling rate: 96 kHz, FFT length: 1024). Acoustic localization was used to identify which member of the dyad produced each whistle. The caller (Delta or Reese) is noted below each whistle and the success sound is identified. Amplitude intensity (dB) shown in the colour bar next to each figure.

bottlenose dolphins [47,48]. The male dyad produced more whistle exchanges than the female dyad, with the males exchanging whistles in 85 trials, compared with just five trials for the females. Interestingly, for the male dyad there was a significant positive correlation between the trial phases (which increased in difficulty) and the proportion of trials that contained whistle exchanges ($N = 8$, $r = 0.74$, $p = 0.02$; figure 4), with either male likely to initiate the exchange (Reese = 45, Delta = 40). Throughout the study, the dolphins used a mixture of signature whistles (i.e. individual identity signals [43]) and other whistle types both within and across trials. Of the 1129 whistles produced by the male dyad across all trial phases; 263 (23%) were Reese's signature whistle and 106 (10%) were Delta's signature whistle. Of the 177 whistles produced by the female dyad across all trial phases; 54 (30%) were Calusa's signature whistle and 17 (10%) were Aleta's signature whistle.

## 4. Discussion

The results show that the two bottlenose dolphin dyads used in this study are capable of using vocal signals to facilitate the successful execution of coordinated, cooperative actions. Given our small sample size, it is not possible to generalize our findings beyond the two dyads. However, our results clearly show that at least some bottlenose dolphins have the cognitive capacity to use communication to actively coordinate their behaviour in a cooperative context. Our two bottlenose dolphin dyads communicated with vocal signals to coordinate their behaviour, with whistle production leading to significantly shorter button press intervals, thereby resulting in more successful trials. Interestingly, the timing of the whistles appeared to be important in facilitating task success. The dolphin dyads produced whistles prior to the button press and probably used these whistles to coordinate when to push, as they were significantly more likely to succeed if they pushed their button after the last

whistle, rather than pushing independently of whistle production. This suggests that whistles can be used as an effective signal to coordinate button pressing.

Although whistles led to increased success for both dyads, their behavioural strategies did differ. The female dyad tended to favour physical synchrony, i.e. waiting nearby for their partner and swimming together when approaching the buttons, and their whistle rates tended to be lower. Given their proximity to each other, whistles are unlikely to be needed to signal general location but can be used to coordinate button presses. Members of the male dyad, on the other hand, tended to move away to other areas of the lagoon until their partner was released and then coordinate at the buttons. This may explain why whistle rates were higher for the male dyad, particularly towards the end of the trials, when they were coordinating at the buttons, and why a higher frequency of whistles also led to significantly more successful trials. By producing more whistles, uncertainty regarding their current locations in the lagoon was reduced, allowing them to locate each other and coordinate their button presses. There was a significant increase in the proportion of vocal exchanges as the male dyad's ability to monitor their partner's behaviour and location became more challenging (i.e. longer delay times between partner's release, and buttons placed further apart).

While the dolphins were significantly more likely to be successful when they used whistles, it is important to note that they were also capable of task success without whistling. In these cases, they were probably coordinating visually or using passive acoustic information as a cue for when to press. Our study was specifically designed so that it was not possible for the dolphins to react to general cues, such as 'press when my partner is near the apparatus' [14]. However, the dolphins do echolocate on the button when pressing it, which takes the form of a short buzz (i.e. rapid click production that can be seen in figure 4). It is conceivable they could have used their partner's echolocation buzz as a cue for when to press if conditions allowed (i.e. low ambient noise and each partner already sufficiently close to their button). Nevertheless, button press intervals were significantly shorter, and the chance of success significantly higher, when active communicative signals, such as whistles, were produced.

In previous studies, great apes have been shown to communicate with gestures to facilitate cooperative task success [33,34], and peach-fronted conures increased their vocalizations during trials where they could not see their partner, although it remains to be determined exactly how the vocalizations were being used [27]. Here, we provide clear evidence that the timing of vocal signals is important for the successful execution of coordinated actions by these two bottlenose dolphin dyads. It is unsurprising that members of some species such as humans and dolphins, and perhaps parrots, favour vocal communication in cooperative tasks: they are among a select number of species capable of diversifying their vocal repertoire through vocal production learning, i.e. they are able to learn novel calls [49]. As such, animals known for their vocal flexibility may have a propensity for using vocal signals to coordinate their actions, or at least have the ability to use that option should the situation require it. While some vocal learners tested using the cooperative rope pulling paradigm remained silent (African grey parrots [31]; Asian elephants [23]), they were always in visual contact and may thus not have needed to engage in vocal coordination.

Extrapolating across studies, it appears that a species' socio-ecology may influence whether they are capable of understanding the need for a partner in a cooperative task [24,27,50]. Bottlenose dolphins cooperate extensively in the wild in the formation of alliances, where males work together to herd single oestrous females and defend them from rival alliances [5,51]. These alliance relationships can last for decades, and are critical to each male's reproductive success [5]. Effective group coordination during these herding events is therefore a key determinant of male fitness, with acoustic coordination playing an important role during this behaviour [38,52]. However, it remains to be seen exactly how male dolphins use vocal signals to coordinate their actions when cooperating in the pursuit and defence of females.

In human societies, it is believed that language, a low-cost, generally honest communication of virtually unlimited complexity, played a role in the widespread radiation of cooperation [32,53]. Indeed, human children have been shown to adapt to complex coordination problems by communicating with vocal signals, leading some scientists to posit that humans evolved unique communication skills in the context of risky coordination problems [35]. The current study shows that at least some bottlenose dolphins can use vocal signals to coordinate their cooperative endeavours, therefore providing compelling new evidence that the flexible use of vocal signals to facilitate cooperative success is not limited to humans.

Ethics. An internal committee at Dolphin Research Center ethically approved this research. All aspects of animal care and habitat complied with the Standards and Guidelines of the Alliance of Marine Mammal Parks and Aquariums. Research was non-invasive and strictly adhered to the laws of the United States of America. The University of Bristol granted animal ethics approvals.

Data accessibility. The dataset supporting this article has been uploaded as part of the electronic supplementary material. Example movies are available from the Dryad Digital Repository: https://doi.org/10.5061/dryad.931zcrjjm [39].

Authors' contributions. S.L.K. and K.J. conceived the study, designed the study, conducted the analysis and drafted the manuscript; E.G., K.D. and C.M. designed the study, conducted the training, trials and behavioral coding; all authors edited the manuscript, provided critical review and gave final approval for submission.

Competing interests. We declare we have no competing interests.

Funding. S.L.K. was supported by The Branco Weiss Fellowship – Society in Science. This work was supported by a research grant from Jim and Marjorie Sanger to Dolphin Research Center.

Acknowledgements. We wish to thank Jesse Fox, Ted Due, Mandy Rodriguez and Jane Hecksher for apparatus design, creation and troubleshooting. We thank Simon J. Allen for providing valuable comments on this manuscript.

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
