## [Peer Review File · Royal Society Open Science]

Review History

RSOS-202073.R0 (Original submission)

Review form: Reviewer 1

Is the manuscript scientifically sound in its present form?

Yes

Are the interpretations and conclusions justified by the results?

Yes

Is the language acceptable?

Yes

Do you have any ethical concerns with this paper?

No

Have you any concerns about statistical analyses in this paper?

No

Recommendation?

Accept with minor revision (please list in comments)

Comments to the Author(s)

General comment: The authors have done a nice job addressing concerns about the use of statistics and mixed models to test whether the dolphins included in this study used vocalizations to coordinate cooperative action during an experimental task. They have been explicit about the statistics representing data for these two dyads, and it is fair for them to state that the results indicate that "at least some dolphins are capable of using vocal signals to facilitate . . . cooperative actions." These modifications do not detract from the value of the paper, which reports very interesting results.

I have a few, very minor suggestions below.

(1) Line 96, Cooperative task methods: I suggest the description and justification of trial phases (currently starting on Line 111) either be placed at the beginning of the "Cooperative task" section, or just start a new paragraph with that sentence to make the description of training phases more obvious to readers.

(2) Results: reporting the sample size

2i: Line 216: You indicate that "Given that sample sizes were limited to two dyads, . . ."

That might be better phrased: "Given that sample sizes represent repeated trials with two dyads, we are unable to use inferential statistics to generalize our findings beyond the dyads. (You can probably simplify this sentence to ". . . we are unable to generalize our findings beyond the dyads", for you are being sufficiently explicit between this and the following sentence.)

If you don't mention the repeated trials in the above sentence, please provide a clear indication of what the sample size, n , represents elsewhere in the 'Statistical analysis' section. That could be placed as a last sentence in this section, e.g., "The sample size refers to the number of replicated trials with each dyad." If you make it clear at the start of this section, this is not necessary.

2ii. When reporting the results in the figures, it would be useful for readers to have a clear indication of what the n represents. For example, in the caption for Figure 2, you can indicate: 'The sample size represents the number of trials per dyad with or without whistles.', with a similar description in other figure captions.

I'm sure the nature of the sample seems obvious to the authors. But readers may be confused at the large apparent sample. I think it is best to be up front – as you have been when revising many of the descriptions in the text. People often look at figures before reading other parts of a paper. They will probably respond more receptively if the nature of the sample is clear.

Susan Lingle

Review form: Reviewer 2

Is the manuscript scientifically sound in its present form?

No

Are the interpretations and conclusions justified by the results?

No

Is the language acceptable?

Yes

Do you have any ethical concerns with this paper?

No

Have you any concerns about statistical analyses in this paper?

Yes

Recommendation?

Major revision is needed (please make suggestions in comments)

Comments to the Author(s)

In this revised version of the manuscript the authors made an effort to acknowledge that their study was not generalizable to all dolphins. The topic remains of interest but could be introduced better (see details below) which would strengthen the paper. The use of statistics is incorrect given the data even after the modification made here. The paper can therefore not be published before all the statistical analyses have been removed.

Introduction

I still feel that the introduction is really short (around 500 words) and only introduce the topic of the paper with 1 single sentence without even a reference. I feel that the paper would be much more appealing for a broad audience if the topic would be introduced, specifically by giving example of coordinated actions in wild animals which are plentiful (e.g. cooperative hunting, cooperative territorial defense, coordinated mate-guarding) and can be found in several taxa including dolphins (e.g. marine mammals, carnivores, apes, fish). This would only require an introductory paragraph and would strengthen the paper. I feel it is particularly important since the results reported here, only based on a sample size of 2 dyads, are anecdotal by nature.

The introduction also does not comprise clear examples of situations in the wild during which dolphins would need to coordinate actions. Such examples are provided in the discussion but should already appear here to better highlight the relevance of the study.

Methods

I appreciate the authors' effort to mention specifically that inferential statistic could not be used here. The question remains why they did statistical analyses at all. As suggested by the editor and myself they should only descriptively explore the data. The use of statistics here is misleading since the model used are statistically incorrect. Using each trial as a single data point is incorrect since the same dyad is tested a large number of time, and this cannot be accounted for by a random effect since there is not enough dyads. Therefore, the statistics presented here are a clear case of pseudo replication and the results cannot be trusted. The statistics should be removed from the paper otherwise that would lead to the publication of statistically incorrect results which is surely undesirable. The results should be presented descriptively, all the stars removed from the graphs and P-values removed from the text.

Discussion

Line 389-393: these type of information would reinforce the introduction greatly.

Decision letter (RSOS-202073.R0)

Dear Dr King

The Editors assigned to your paper RSOS-202073 "Bottlenose dolphins can communicate with vocal signals to solve a cooperative task" have now received comments from reviewers and would like you to revise the paper in accordance with the reviewer comments and any comments from the Editors. Please note this decision does not guarantee eventual acceptance.

Please submit your revised manuscript and required files (see below) no later than 21 days from today's (ie 25-Jan-2021) date. Note: the ScholarOne system will 'lock' if submission of the revision is attempted 21 or more days after the deadline. If you do not think you will be able to meet this deadline please contact the editorial office immediately.

on behalf of Professor Kevin Padian (Subject Editor)
openscience@royalsociety.org

Associate Editor Comments to Author:

It is clear that you have made substantial efforts to resolve the concerns from the earlier review at our sister journal PRSB. However, it is also plain that a number of matters remain to be resolved - the second reviewer here has identified several concerns regarding, for instance, the statistical treatment of the study. These will need to be addressed before the paper may be considered

ready for publication - please ensure that you fully address these matters as it is unlikely that a further round of revision will be permitted.

Reviewer comments to Author:

Reviewer: 1

Comments to the Author(s)

General comment: The authors have done a nice job addressing concerns about the use of statistics and mixed models to test whether the dolphins included in this study used vocalizations to coordinate cooperative action during an experimental task. They have been explicit about the statistics representing data for these two dyads, and it is fair for them to state that the results indicate that "at least some dolphins are capable of using vocal signals to facilitate . . . cooperative actions." These modifications do not detract from the value of the paper, which reports very interesting results.

I have a few, very minor suggestions below.

(1) Line 96, Cooperative task methods: I suggest the description and justification of trial phases (currently starting on Line 111) either be placed at the beginning of the "Cooperative task" section, or just start a new paragraph with that sentence to make the description of training phases more obvious to readers.

(2) Results: reporting the sample size

2i: Line 216: You indicate that "Given that sample sizes were limited to two dyads, . . ."

That might be better phrased: "Given that sample sizes represent repeated trials with two dyads, we are unable to use inferential statistics to generalize our findings beyond the dyads. (You can probably simplify this sentence to ". . . we are unable to generalize our findings beyond the dyads", for you are being sufficiently explicit between this and the following sentence.)

If you don't mention the repeated trials in the above sentence, please provide a clear indication of what the sample size, n , represents elsewhere in the 'Statistical analysis' section. That could be placed as a last sentence in this section, e.g., "The sample size refers to the number of replicated trials with each dyad." If you make it clear at the start of this section, this is not necessary.

2ii. When reporting the results in the figures, it would be useful for readers to have a clear indication of what the n represents. For example, in the caption for Figure 2, you can indicate: 'The sample size represents the number of trials per dyad with or without whistles.', with a similar description in other figure captions.

I'm sure the nature of the sample seems obvious to the authors. But readers may be confused at the large apparent sample. I think it is best to be up front - as you have been when revising many of the descriptions in the text. People often look at figures before reading other parts of a paper. They will probably respond more receptively if the nature of the sample is clear.

Susan Lingle

Reviewer: 2
 Comments to the Author(s)

In this revised version of the manuscript the authors made an effort to acknowledge that their study was not generalizable to all dolphins. The topic remains of interest but could be introduced better (see details below) which would strengthen the paper. The use of statistics is incorrect given the data even after the modification made here. The paper can therefore not be published before all the statistical analyses have been removed.

Introduction

I still feel that the introduction is really short (around 500 words) and only introduce the topic of the paper with 1 single sentence without even a reference. I feel that the paper would be much more appealing for a broad audience if the topic would be introduced, specifically by giving example of coordinated actions in wild animals which are plentiful (e.g. cooperative hunting, cooperative territorial defense, coordinated mate-guarding) and can be found in several taxa including dolphins (e.g. marine mammals, carnivores, apes, fish). This would only require an introductory paragraph and would strengthen the paper. I feel it is particularly important since the results reported here, only based on a sample size of 2 dyads, are anecdotal by nature.

The introduction also does not comprise clear examples of situations in the wild during which dolphins would need to coordinate actions. Such examples are provided in the discussion but should already appear here to better highlight the relevance of the study.

Methods

I appreciate the authors' effort to mention specifically that inferential statistic could not be used here. The question remains why they did statistical analyses at all. As suggested by the editor and myself they should only descriptively explore the data. The use of statistics here is misleading since the model used are statistically incorrect. Using each trial as a single data point is incorrect since the same dyad is tested a large number of time, and this cannot be accounted for by a random effect since there is not enough dyads. Therefore, the statistics presented here are a clear case of pseudo replication and the results cannot be trusted. The statistics should be removed from the paper otherwise that would lead to the publication of statistically incorrect results which is surely undesirable. The results should be presented descriptively, all the stars removed from the graphs and P-values removed from the text.

Discussion

Line 389-393: these type of information would reinforce the introduction greatly.

===PREPARING YOUR MANUSCRIPT===

Your revised paper should include the changes requested by the referees and Editors of your manuscript. You should provide two versions of this manuscript and both versions must be provided in an editable format:
 one version identifying all the changes that have been made (for instance, in coloured highlight, in bold text, or tracked changes);
 a 'clean' version of the new manuscript that incorporates the changes made, but does not highlight them. This version will be used for typesetting if your manuscript is accepted.
 Please ensure that any equations included in the paper are editable text and not embedded images.

Please ensure that you include an acknowledgements' section before your reference list/bibliography. This should acknowledge anyone who assisted with your work, but does not

qualify as an author per the guidelines at <https://royalsociety.org/journals/ethics-policies/openness/>.

===PREPARING YOUR REVISION IN SCHOLARONE===

- Ensure that your data access statement meets the requirements at <https://royalsociety.org/journals/authors/author-guidelines/#data>. You should ensure that you cite the dataset in your reference list. If you have deposited data etc in the Dryad repository, please include both the 'For publication' link and 'For review' link at this stage.
- If you are requesting an article processing charge waiver, you must select the relevant waiver option (if requesting a discretionary waiver, the form should have been uploaded at Step 3 'File upload' above).
- If you have uploaded ESM files, please ensure you follow the guidance at <https://royalsociety.org/journals/authors/author-guidelines/#supplementary-material> to include a suitable title and informative caption. An example of appropriate titling and captioning may be found at https://figshare.com/articles/Table_S2_from_Is_there_a_trade-off_between_peak_performance_and_performance_breadth_across_temperatures_for_aerobic_scope_in_teleost_fishes_/3843624.

Author's Response to Decision Letter for (RSOS-202073.R0)

See Appendix A.

RSOS-202073.R1 (Revision)

Review form: Reviewer 2

Is the manuscript scientifically sound in its present form?

Yes

Are the interpretations and conclusions justified by the results?

Yes

Is the language acceptable?

Yes

Do you have any ethical concerns with this paper?

No

Have you any concerns about statistical analyses in this paper?

No

Recommendation?

Accept as is

Comments to the Author(s)

The authors have thoroughly revised their interpretation of the results and the manuscript can be published as such.

Decision letter (RSOS-202073.R1)

Dear Dr King,

It is a pleasure to accept your manuscript entitled "Evidence that bottlenose dolphins can communicate with vocal signals to solve a cooperative task" in its current form for publication in Royal Society Open Science. The comments of the reviewers who reviewed your manuscript are included at the foot of this letter.

on behalf of Professor Kevin Padian (Subject Editor)
openscience@royalsociety.org

Associate Editor Comments to Author:

The reviewer considers your work ready for acceptance and publication - congratulations and thank you for the support of the journal! We hope you will submit to us again in future (and we certainly encourage it!).

Reviewer comments to Author:

Reviewer: 2

Comments to the Author(s)

The authors have thoroughly revised their interpretation of the results and the manuscript can be published as such.

Appendix A

Associate Editor Comments to Author:

It is clear that you have made substantial efforts to resolve the concerns from the earlier review at our sister journal PRSB. However, it is also plain that a number of matters remain to be resolved - the second reviewer here has identified several concerns regarding, for instance, the statistical treatment of the study. These will need to be addressed before the paper may be considered ready for publication - please ensure that you fully address these matters as it is unlikely that a further round of revision will be permitted.

Author response: We have now revised the manuscript further in light of these comments. We provide a detailed point by point response to the reviewer's concerns below. With regards to the comment from Reviewer 2 regarding the statistical treatment of the study, we provide a comprehensive answer to that concern below. To summarise, if the scope of the question is understanding responses only at the level they were measured i.e., the dyad, then inferential statistics are valid for that purpose (we provide recent papers on this topic) – we are just not able to generalize our findings beyond the dyads. We have, thus, retained the statistics but revised our manuscript further to make it explicitly clear that our results only pertain to these two dyads. This includes toning down the title and using due caution throughout the manuscript (Abstract, lines: 25-26 and 30-33; Methods, lines 26-29; Results, lines 273-274, Discussion, lines 346-348, 409-410, and 416-417).

Reviewer: 1

General comment: The authors have done a nice job addressing concerns about the use of statistics and mixed models to test whether the dolphins included in this study used vocalizations to coordinate cooperative action during an experimental task. They have been explicit about the statistics representing data for these two dyads, and it is fair for them to state that the results indicate that “at least some dolphins are capable of using vocal signals to facilitate . . . cooperative actions.” These modifications do not detract from the value of the paper, which reports very interesting results.

Author response: We thank the reviewer for their positive appraisal of our manuscript.

I have a few, very minor suggestions below.

(1) Line 96, Cooperative task methods: I suggest the description and justification of trial phases (currently starting on Line 111) either be placed at the beginning of the “Cooperative task” section, or just start a new paragraph with that sentence to make the description of training phases more obvious to readers.

Author response: We have now made this a new paragraph.

(2) Results: reporting the sample size

2i: Line 216: You indicate that “Given that sample sizes were limited to two dyads, . . .”

That might be better phrased: “Given that sample sizes represent repeated trials with two dyads, we are unable to use inferential statistics to generalize our findings beyond the dyads.

(You can probably simplify this sentence to “. . . we are unable to generalize our findings beyond the dyads”, for you are being sufficiently explicit between this and the following sentence.)

If you don't mention the repeated trials in the above sentence, please provide a clear indication of what the sample size, n , represents elsewhere in the 'Statistical analysis' section. That could be placed as a last sentence in this section, e.g., “The sample size refers to the number of replicated trials with each dyad.” If you make it clear at the start of this section, this is not necessary.

Author response: We have made the initial suggested change. The sentence now reads: “Given that sample sizes represent repeated trials with two dyads, we are unable to generalize our findings beyond the dyads.”

We have also clarified throughout this section that the tests we are using are exploring the behaviour “for these two dyads”.

2ii. When reporting the results in the figures, it would be useful for readers to have a clear indication of what the n represents. For example, in the caption for Figure 2, you can indicate: ‘The sample size represents the number of trials per dyad with or without whistles.’, with a similar description in other figure captions.

I'm sure the nature of the sample seems obvious to the authors. But readers may be confused at the large apparent sample. I think it is best to be up front – as you have been when revising many of the descriptions in the text. People often look at figures before reading other parts of a paper. They will probably respond more receptively if the nature of the sample is clear.

Author response: This is a good point and one we previously overlooked. We have now added “Sample size (N) represents the number of trials per dyad with or without whistles.” To the Figure 2 legend.

Susan Lingle

Reviewer: 2

Comments to the Author(s)

In this revised version of the manuscript the authors made an effort to acknowledge that their study was not generalizable to all dolphins. The topic remains of interest but could be introduced better (see details below) which would strengthen the paper. The use of statistics is incorrect given the data even after the modification made here. The paper can therefore not be published before all the statistical analyses have been removed.

Author response: Thank you for the constructive comments on our paper. However, the use of statistics is not incorrect. Below the reviewer states that “The use of statistics here is misleading since the model used are statistically incorrect. Using each trial as a single data point is incorrect since the same dyad is tested a large number of time, and this cannot be accounted for by a random effect since there is not enough dyads. Therefore, the statistics presented here are a clear case of pseudo replication and the results cannot be trusted”

The risk of type 1 errors with pseudo-replication is at the level of the population or species. If the scope of the question is understanding responses only at the level they were measured i.e., the dyad, then inferential statistics can be used to examine the extent that observed differences are meaningful for that individual/dyad etc. The reviewer is absolutely correct that we cannot generalize beyond our two dyads, and we have made further revisions to the manuscript to ensure that our results cannot be misconstrued by the reader (including toning down the title). We hope the reviewer finds the manuscript much improved as a result. However, we have not removed the statistics from the paper because the inclusion of these statistical tests to quantify the behaviour of our two dyads is valid. We cite recent papers on this topic below.

1. C. Y. Jordan, Population sampling affects pseudoreplication. *PLoS Biol.* **16**, 1–3 (2018).

For example, see Page 2/3 (Similarly, if a study uses only samples from a single individual, genotype, or family, it will be pseudoreplicated if it aims to make inferences at a higher level (e.g., for a population or species) but not if the study aims to understand responses only at the level at which they are measured: whether a study is pseudoreplicated depends on its scope).

2. S. E. Lazic, C. J. Clarke-Williams, M. R. Munafò, What exactly is “N” in cell culture and animal experiments? *PLOS Biol.* **16**, e2005282 (2018).

For example, see Page 10/14 (Nonhuman primate experiments tend to have small sample sizes for both cost and ethical reasons, but the requirements for genuine replication remain the same. Suppose an experiment records from a single neuron while pictures of a happy monkey or a sad monkey are shown to the subject. One-hundred pictures are shown in random order, and for each trial the firing rate of the neuron is measured. We find that the neuron fires at a faster rate in the happy monkey condition, and the sample size is the 100 trials (this design resembles Fig 2D). This seems to be an easy way of obtaining a large sample size with only 1 subject and only 1 neuron, but the catch is that the results apply only to this subject and to this neuron, and little can be said about what might be seen in other subjects or other neurons. A statistical test would be valid, but the hypothesis tested is uninteresting. One may argue that this subject is representative of others, but this is a nonstatistical generalisation, and the smallness of the p-value does not provide more evidence about what might happen in other subjects (showing that a drug works for Jim ($p < 0.001$) does not provide strong evidence that it works for Bob, or anyone else).

3. N. Colegrave, G. D. Ruxton, Using Biological Insight and Pragmatism When Thinking about Pseudoreplication. *Trends Ecol. Evol.* **33**, 28–35 (2018).

For example, see Box 2 (Our view is that such use of statistics can aid the reader and should not mislead the reader provided the authors stick to interpreting their data appropriately, essentially remembering that they are seeking to understand one specific island and not islands generally).

Introduction

I still feel that the introduction is really short (around 500 words) and only introduce the topic of the paper with 1 single sentence without even a reference. I feel that the paper would be much more appealing for a broad audience if the topic would be introduced, specifically by giving example of coordinated actions in wild animals which are plentiful (e.g. cooperative hunting, cooperative territorial defense, coordinated mate-guarding) and can be found in several taxa including dolphins (e.g. marine mammals, carnivores, apes, fish). This would only require an introductory paragraph and would strengthen the paper. I feel it is particularly important since the results reported here, only based on a sample size of 2 dyads, are anecdotal by nature.

The introduction also does not comprise clear examples of situations in the wild during which

dolphins would need to coordinate actions. Such examples are provided in the discussion but should already appear here to better highlight the relevance of the study.

Author response: We have now revised our introduction to include a discussion of coordinated actions in other species and then specific examples of cooperation/coordination in wild dolphins.

Methods

I appreciate the authors' effort to mention specifically that inferential statistic could not be used here. The question remains why they did statistical analyses at all. As suggested by the editor and myself they should only descriptively explore the data. The use of statistics here is misleading since the model used are statistically incorrect. Using each trial as a single data point is incorrect since the same dyad is tested a large number of time, and this cannot be accounted for by a random effect since there is not enough dyads. Therefore, the statistics presented here are a clear case of pseudo replication and the results cannot be trusted. The statistics should be removed from the paper otherwise that would lead to the publication of statistically incorrect results which is surely undesirable. The results should be presented descriptively, all the stars removed from the graphs and P-values removed from the text.

Author response: The statistics are not incorrect, nor are the results themselves incorrect (see response above), but the generalization of our results would be incorrect. We have taken further measures to revise the manuscript, making it explicitly clear we cannot generalize beyond our two dyads.

Discussion

Line 389-393: these type of information would reinforce the introduction greatly.

Author response: We have now revised the introduction accordingly.